# Ephedrae Herba and Cinnamomi Cortex interactions with G glycoprotein inhibit respiratory syncytial virus infectivity

Aya Fujikane [1,2], Atsuhiko Sakamoto[1,2], Ryosuke Fujikane [3,4], Akinori Nishi[5], Yoshizumi Ishino [6], Kenji Hiromatsu[2] & Shigeki Nabeshima [1✉]

Although respiratory syncytial virus (RSV) is a major cause of respiratory tract infection in children, no effective therapies are available. Recently, RSV G, the attachment glycoprotein, has become a major focus in the development of therapeutic strategies against RSV infection. Treatment of RSV-infected cultured cells with maoto, a traditional herbal medicine for acute febrile diseases, significantly reduced the viral RNA and titers. RSV attachment to the cell surface was inhibited both in the presence of maoto and when RSV particles were pre-treated with maoto. We demonstrated that maoto components, Ephedrae Herba (EH) and Cinnamomi Cortex (CC), specifically interacted with the central conserved domain (CCD) of G protein, and also found that this interaction blocked viral attachment to the cellular receptor CX3CR1. Genetic mutation of CX3C motif on the CCD, the epitope for CX3CR1, decreased the binding capacity to EH and CC, suggesting that CX3C motif was the target for EH and CC. Finally, oral administration of maoto for five days to RSV-infected mice significantly reduced the lung viral titers. These experiments clearly showed the anti-RSV activity of EH and CC mixed in maoto. Taken together, this study provides insights for the rational design of therapies against RSV infection.

[1] General Medicine, Fukuoka University Hospital, Fukuoka, Japan. [2] Department of Microbiology and Immunology, Faculty of Medicine, Fukuoka University, Fukuoka, Japan. [3] Department of Physiological Science and Molecular Biology, Fukuoka Dental College, Fukuoka, Japan. [4] Oral Medicine Research Center, Fukuoka Dental College, Fukuoka, Japan. [5] Tsumura Kampo Research Laboratories, Tsumura & Co., Ibaraki, Japan. [6] Department of Bioscience and Biotechnology, Graduate School of Bioresource and Bioenvironmental Sciences, Kyushu University, Fukuoka, Japan. ✉email: snabeshi@fukuoka-u.ac.jp

Respiratory syncytial virus (RSV) causes acute respiratory tract infection in individuals of all ages. Infection during infancy is especially common, and sometimes leads to hospitalization and death[1,2]. Despite decades of efforts, no effective therapies or vaccines are available[3–5]. Prophylaxis with anti-RSV antibodies, palivizumab, is limited to the high-risk children for preventing serious disease. At present, supportive care remains the only way for patients with RSV infection. Control of RSV infection is an urgent global problem, and the development of new antiviral drug is highly expected.

RSV belongs to the Orthopneumovirus genus of the Pneumoviridae family and has a negative-sense, single-stranded RNA genome. The viral envelope contains two major glycoproteins[6]: the G glycoprotein, which mediates attachment to CX3C chemokine receptor 1 (CX3CR1) on the cell surface[7–10], and the F glycoprotein, which mediates fusion by interacting with the insulin-like growth factor 1 receptor and the coreceptor nucleolin[11,12]. The G protein is more sequence diverse than F, except for a central conserved domain (CCD) that is nearly invariant across RSV strains[13]. Recently, the RSV G has become a major focus in the development of prophylactic and therapeutic strategies against RSV infection, with particular emphasis on the CCD because this domain contains the major epitope involved in cell surface attachment[14,15]. The CCD bears a conformationally constrained CX3C chemokine motif that is thought to be important for binding to CX3CR1 and chemokine secretion[16,17]. Previous reports showed that RSV infection of mice or cultured lung cells could be inhibited by blockade of the CCD-CX3CR1 interaction using monoclonal antibodies against G protein or by genetic mutations of the CCD[18–24].

Traditional herbal medicines have long played important roles in countries of the Far East, especially Japan, China, and Korea. Traditional herbal medicine, called Kampo, is accepted by the national medical insurance system of Japan, allowing the widespread use of these medicines by physicians. Maoto (ma-huang-tang in Chinese) has shown clinical efficacy for treatment of common cold and influenza symptoms. Maoto is a multicomponent formulation extracted from four plants: Ephedrae Herba (EH), Cinnamomi Cortex (CC), Armeniacae Semen (AS), and Glycyrrhize Radix (GR) (Supplementary Table 1). The formulation contains over 350 compounds; the major active compounds are ephedrine, methylephedrine, hippuric acid, and glycyrrhetinic acid[25,26]. We previously reported the results of two clinical trials showing that maoto was tolerable and effective in treating seasonal influenza by comparison with neuraminidase inhibitors[27,28]. Using our in vitro infection system, we previously showed that EH and CC in maoto caused influenza virus particles to remain in endosomes because of a failure in viral fusion with the endosome membrane through the elevation of endosomal pH condition[29].

We report here a significant reduction of viral loads in RSV-infected A549 cells treated with maoto. We investigated the underlying mechanism and found that maoto components, especially EH and CC, interacted with the CX3C chemokine motif on the CCD of RSV G protein and blocked attachment to cellular CX3CR1. These in vitro results were confirmed by in vivo analysis, that oral administration of maoto to mice infected with RSV showed the anti-viral and anti-inflammatory effect in lung. Taken together, the present study provides insights into the treatment of RSV infection. Since maoto is a clinically proven drug, further clinical application is feasible compared with newly designed antivirals which take high costs and limited administration routes.

## Results

### Inhibitory effects of maoto in cultured cells infected with RSV.
We first assessed the antiviral activity of maoto in A549 cells infected with RSV (subtype A2) by real-time (RT)-PCR to quantitate intracellular viral RNA, as we did in previous experiments for influenza virus[29]. To avoid measuring progeny virus RNA, culture time was limited to 6 h. Because the viral life cycle consists of attachment of virions to the cell surface followed by fusion, entry and subsequent replication, we divided the life cycle into two phases: the binding phase (first 1 h post infection) and the entry/replication phase (subsequent 6-h period after washing out residual virions)[30]. We found that viral RNA levels were remarkably decreased when maoto was present at the binding phase, whereas no significant decrease was observed when maoto was present at the entry/replication phase (Fig. 1a). When we pretreated A549 cells with maoto prior to the binding phase, there was no decrease in viral RNA level (Supplementary Fig. 1a), suggesting that host natural immunity or restriction factors with anti-RSV activity were not induced by maoto. On the basis of these results, in subsequent experiments maoto was added only at the binding phase. As shown in Fig. 1b, maoto treatment of A549 cells resulted in decreased RSV viral RNA levels in a dose-dependent manner with an IC50 of 1.77 μg/ml. When cells were infected with 10 times the amount of viruses (moi 10), the IC50 of maoto had changed to 21.57 μg/ml (Supplementary Fig. 2a). Together, these experiments suggested that maoto inhibits viral infectivity in the early phase of the viral life cycle.

A plaque-forming unit (PFU) assay showed significant reductions in the viral titers of culture supernatants treated with maoto at the binding phase. This result indicated that formation of infective progeny viruses was significantly decreased by maoto in a dose-dependent manner (Fig. 1c). We next visualized replicated viral components in host cells by immunofluorescence confocal microscopy. Translated viral G proteins (green fluorescence) were observed in the cytoplasm 24 h post-RSV infection (Supplementary Fig. 1b). The presence of maoto at the binding phase decreased the percentage of RSV-positive cells 24 h post-RSV infection in a dose-dependent manner (Supplementary Fig. 1c).

Because maoto is composed of extracts from four plants (EH, AS, CC, and GR), we next assessed which plants were responsible for inhibition of viral infection. We found that EH and CC, but not AS and GR, showed significant antiviral effects (Fig. 1d). We next asked whether the antiviral effect of maoto was RSV-A2 or host cell-dependent. Treatment of RSV-B-infected A549 cells with maoto significantly reduced viral RNA levels in a dose-dependent manner with an IC50 of 5.70 μg/mL. (Supplementary Fig. 3a). We also confirmed the anti-RSV-A effect of maoto in Vero cells derived from African green monkeys (Supplementary Fig. 3b). These results indicated that maoto acts independently of both viral subtype and host cell line.

### Direct inhibitory effect of maoto on RSV.
Two mechanisms could explain the above results: either maoto interacted with RSV itself or with host cell receptors for RSV. To test potential direct effects of maoto on virions, high-concentration RSV was incubated with maoto for 0.5 min or 60 min. A549 cells were infected with the treated virions at a multiplicity of infection (moi) 1, then cultured for 6 h (maoto concentration <1 μg/mL) (Fig. 2a). We observed a significant decrease in viral RNA level when virions were exposed to maoto for 60 min, suggesting a direct effect of maoto on RSV. To exclude potential effects of maoto on cellular receptors, cells were incubated with maoto for 1 h on ice prior to RSV infection. No decrease in viral RNA level was observed, suggesting that maoto had no effect on cellular receptors for RSV (Supplementary Fig. 4). Therefore, we next asked whether maoto disrupted RSV or interacted with specific RSV epitopes to prevent binding to cells. To distinguish between these possibilities, RSV attached to the cell surface was treated with maoto on ice. At low

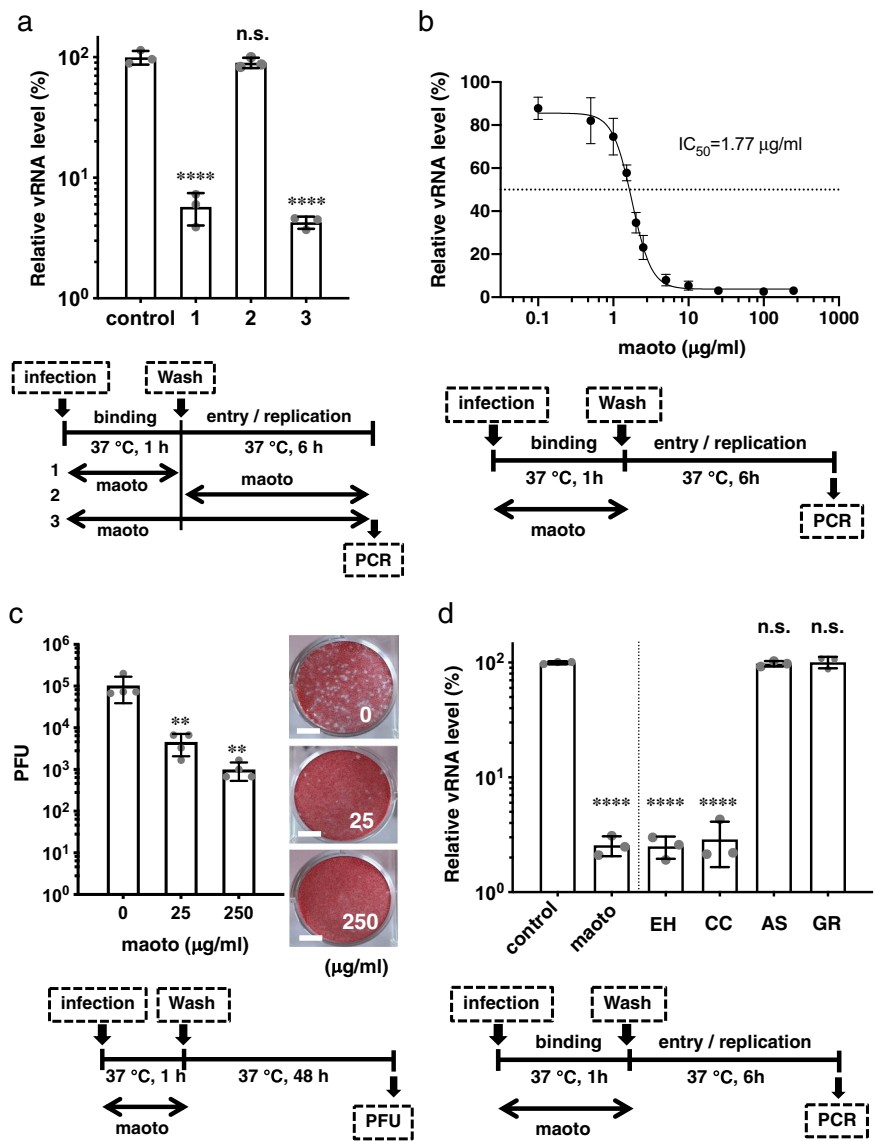

**Fig. 1 Anti-RSV activity of maoto in cultured cells in the early phase of the viral life cycle. a** Anti-RSV activity of maoto in A549 cells. A549 cells were infected with RSV-A (multiplicity of infection [moi] 1) and treated with maoto (25 μg/mL) at the binding phase (1), entry/replication phase (2), and both (3), followed by culture for six hours. Viral RNA (vRNA) in the cells was quantified by RT-PCR. The culture condition was illustrated bellow. **b** Dose-dependent decrease of vRNA by maoto. A measure of IC50 was determined using logistic regression testing. The culture conditions are illustrated bellow. **c** Production of infectious progeny viruses was inhibited by maoto. A549 cells were infected with RSV in the presence or absence of maoto at the binding phase, and then cultured for 48 h. Viral titers in supernatants were determined by plaque forming assay. Scale bars indicate 5 mm. **d** Herbal components of maoto that block the infectivity of RSV. EH; Ephedrae Herba (7.5 μg/mL), CC; Cinnamomi Cortex (7.5 μg/mL), AS; Armeniacae Semen (7.5 μg/mL), GR; Glycyrrhize Radix (2.5 μg/mL). **a**–**d** show individual values and mean±error bars SD (**a**, **d**: $n = 3$, **b**, **c**: $n = 4$). One-way ANOVA followed by Dunnett post-test. **$P < 0.01$, ****$P < 0.0001$, and n.s.; not significant.

temperatures, virions remain bound to cellular receptors without entering the cytoplasm because cellular metabolism is slowed[30]. Following maoto treatment, cells were cultivated at 37 °C for 24 h without maoto. As shown in Fig. 2b, viral RNA was propagated in spite of prior exposure to maoto, indicating that maoto does not directly inactivate RSV but may instead interact with RSV epitopes required for cellular receptor attachment.

To investigate whether maoto could inhibit viral attachment to the cell surface, we further examined the amount of viral RNA and protein located on the cell surface at the binding phase. RSV was allowed to bind to cells in the presence or absence of maoto for 1 h, and then the amount of viral RNA in cell extracts was assayed by RT-qPCR (Fig. 2c). Cell surface viral RNA levels were significantly reduced by maoto treatment. This result was

confirmed at the protein level: cell surface RSV G protein levels were reduced when infection was performed with maoto as shown by western blotting (Fig. 2d). We also show that a higher concentration of maoto is required for inhibiting an increasing number of viruses (at a moi of 10) (Supplementary Fig. 2b). RSV attachment to the cell surface was observed and quantified by immunofluorescence confocal microscopy (Fig. 2e). Virion foci significantly decreased in a dose-dependent manner when cells were treated with maoto (Supplementary Fig. 5).

**Interaction of maoto components with RSV G protein**. We hypothesized that maoto components might interact with viral envelope glycoproteins to inhibit RSV attachment to cellular

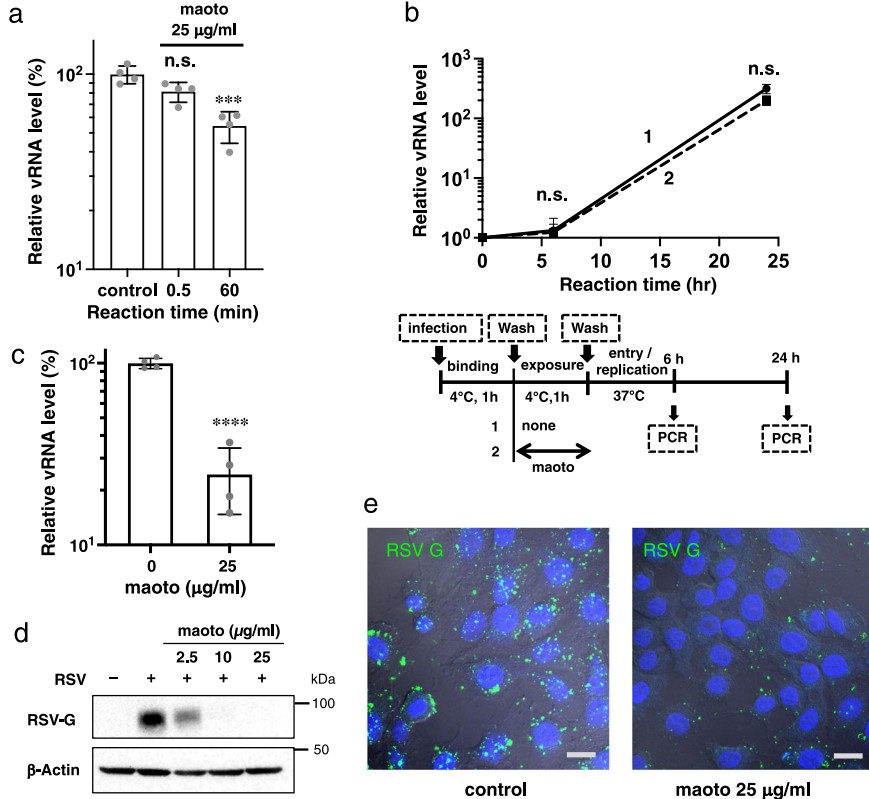

**Fig. 2 Direct inhibitory effect of maoto on RSV. a** Maoto directly inhibits the RSV infectivity. High-concentration RSV was incubated with maoto (25 μg/ mL) for 0.5 or 60 min. A549 cells were inoculated with the treated RSV (moi 1) and cultured for 6 h (maoto concentration 0.625 μg/mL). Levels of viral RNA in the cells were measured by RT-PCR. One-way ANOVA followed by Dunnett's post-test. ***$P < 0.001$ and n.s.; not significant. **b** Maoto does not disrupt RSV. RSV was allowed to attach to cells for 1 h on ice and then incubated for 1 h in the presence or absence of maoto. After washing, cells were cultured at 37 °C for 24 h and viral RNA levels in the cells were measured by RT-PCR. The culture conditions are illustrated below. n.s.; not significant. **c** Reduction of cell-surface viral RNA levels by maoto. A549 cells were inoculated with RSV (moi 1) and simultaneously treated with maoto on ice for 1 h. After washing, viral RNA levels in the cell were measured by RT-PCR. Unpaired Student's *t*-test was used. ****$P < 0.0001$ versus 0 μg/ml. **d** Reduction of cell surface RSV G protein levels by maoto. A549 cells were inoculated with RSV (moi 1) and simultaneously treated with maoto on ice for 1 h. RSV G protein in the cells was detected by western blotting. **e** Visualization of RSV G protein on the cell surface. Merged photos show RSV G protein (green), DAPI (blue), and cells. **a**–**c** show individual values and mean±error bars SD (**a**, **c**: $n = 4$, **b**: $n = 3$). **e**, Scale bars indicate 10 μm.

receptors. We focused on the RSV G protein, which was reported to be the major protein responsible for virion attachment to host cell receptors. To test our hypothesis, the binding of maoto to RSV G protein was qualitatively examined using surface plasmon resonance (SPR). When we injected various concentrations of maoto over G protein-immobilized surfaces, dose-dependent binding was observed (Fig. 3a). After washing (>120 s), the sensorgrams showed the analyte remained on G protein-immobilized surfaces, indicating that maoto hardly dissociates from RSV G protein. The binding response in this SPR analysis may include both specific and non-specific bindings, because maoto is composed of crude extracts. However, in contrast to the case of RSV G protein, maoto did not interact with severe acute respiratory syndrome coronavirus-2 S glycoprotein (Fig. 3b), suggesting that the binding of maoto to RSV G protein was considerably selective. Because maoto components have varying molecular weights, dissociation constants (KD) could not be calculated. We next assessed which maoto components (EH, CC, GR, or AS) interacted with RSV G protein. We found that EH and CC, but not GR and AS, bound RSV G protein (Fig. 3c). This result was consistent with the effects of these components on cultured cells (Fig. 1d). Because the RSV G protein consists of heavily glycosylated mucin-like domains, it shows a broad band in western blotting studies (Fig. 3d, left). We next examined whether maoto targeted proteins or glycans of the RSV G protein. We prepared a non-

glycosylated RSV G protein extracellular domain in *Escherichia coli* (Fig. 3d, right). Dose-dependent binding of maoto to non-glycosylated G protein was observed by SPR (Fig. 3e), suggesting that the maoto reaction with RSV G protein was not glycan-dependent.

We next examined whether the maoto-RSV G protein interaction inhibited the attachment of RSV to cells. As shown in Fig. 3f, CX3CR1, the target for RSV G attachment, was present on A549 cells as shown by immunofluorescence confocal microscopy using a monoclonal anti-CX3CR1 antibody. We investigated the effect of maoto on the interaction between RSV G protein and cellular CX3CR1 using a proximity ligation assay (PLA) which was developed for the detection of protein-protein interaction in situ. RSV was allowed to attach to A549 cells on ice in the presence or absence of maoto, followed by the PLA (Fig. 3g). Many foci reflecting the proximal association of RSV G protein and CX3CR1 were observed on A549 cells in the absence of maoto. However, significantly decreased numbers of foci were observed in cells treated with maoto (Supplementary Fig. 6). These results demonstrated that specific binding of G protein to CX3XR1 was inhibited by maoto.

**Specific interaction of maoto with the CX3C motif on CCD.** The CX3C motif on the CCD of RSV G protein was reported to be a candidate epitope responsible for virion attachment to

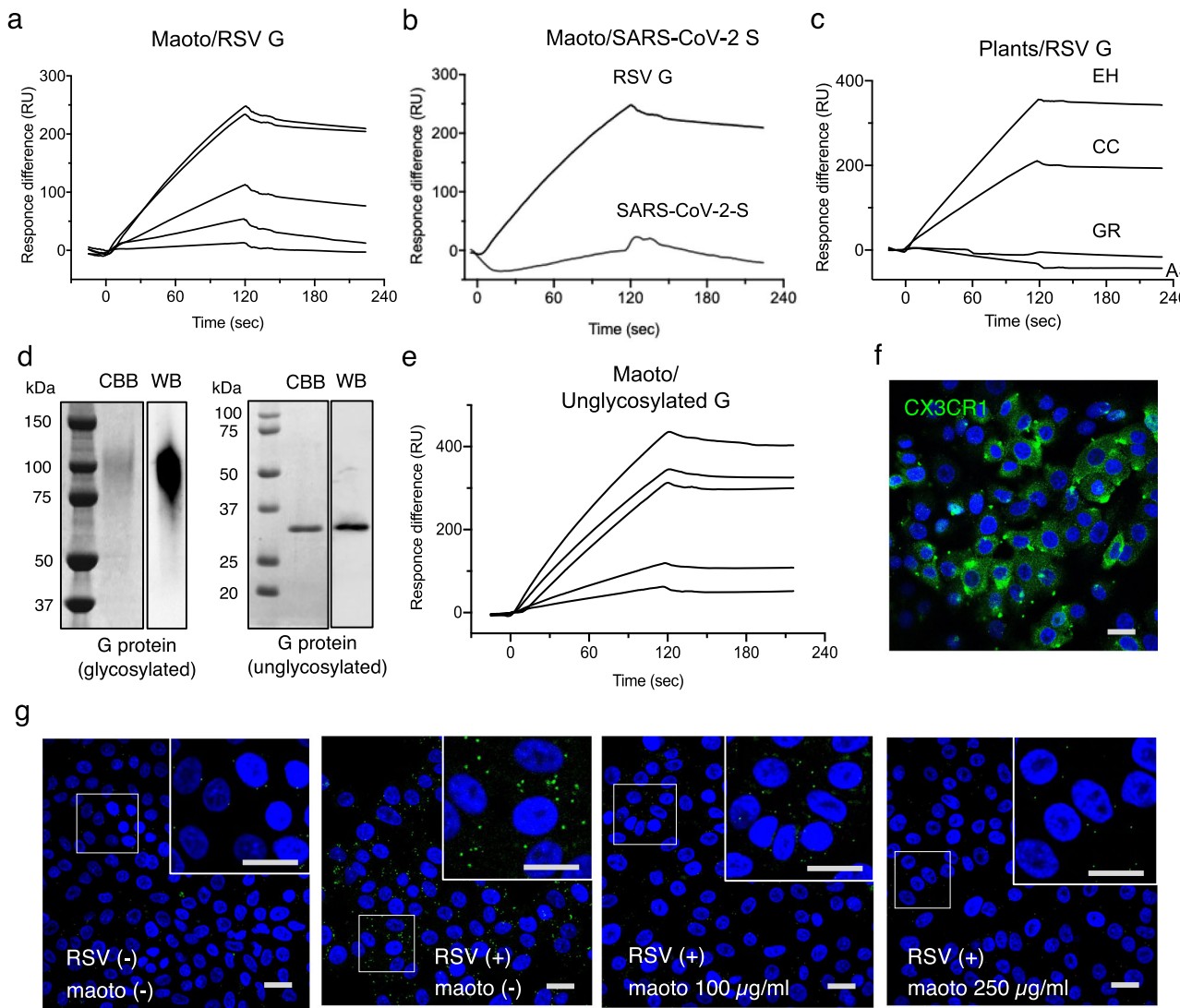

**Fig. 3 Specific binding of maoto to the central conserved domain (CCD) of RSV G glycoprotein. a** Binding of maoto to RSV G protein immobilized on a sensor chip by surface plasmon resonance (SPR). Five different concentrations (5, 10, 20, 30 and 40 μg/mL) of maoto were injected. **b** Selective binding of maoto to RSV G protein. Maoto (40 μg/mL) was injected to severe acute respiratory syndrome coronavirus-2 (SARS-CoV-2) S protein or RSV G protein immobilized on a sensor chip by SPR. **c** Binding of Ephedra Herba (EH), Cinnamoni Cortex (CC), Glycyrrhize Radix (GR), and Armeniacae Semen (AS) to RSV G protein immobilized on a sensor chip by SPR. The concentration of each component was 10 μg/mL. **d** Molecular weights of glycosylated RSV G protein (left panel) and non-glycosylated RSV G protein (right panel). Glycosylated and non-glycosylated G protein were produced in mammalian cells and *Escherichia coli*, respectively. The proteins were detected by CBB: Coomassie brilliant blue or WB: Western blotting. **e** Interaction of maoto with non-glycosylated RSVG protein immobilized on a sensor chip. Binding was assessed by SPR. Five different concentrations (5, 10, 20, 30 and 40 μg/mL) of maoto were injected. **f** Expression of CX3CR1 on A549 cells visualized by confocal microscopy (green foci). The horizontal bar indicates 20 μm. **g** Inhibition of RSV attachment to CX3CR1 by maoto. RSV (moi 10) was allowed to attach to A549 cells on ice in the presence or absence of maoto. Binding of RSV G protein to CX3CR1 was examined using a proximity ligation assay (PLA). Green foci on cultured cells indicate the PLA probes. **f**, **g** Scale bars indicate 20 μm.

CX3CR1[17]. Therefore, we next examined whether maoto could interact directly with the CCD CX3C motif. We synthesized a peptide derived from the CCD (CX3C, aa 164–186) and a mutant peptide in the CX3C motif (SX3S), the latter was replaced cysteines of CX3C motif into serines (Fig. 4a)[9]. SPR revealed strong dose-dependent binding of maoto to the CX3C peptide (Fig. 4b), but around 50% weaker binding to the SX3S peptide (Fig. 4c). Both EH and CC bound to the two peptides similarly to maoto (Fig. 4d, e). These results demonstrated that maoto and its components (EH and CC) preferentially bind the CX3C motif on the RSV G protein CCD to inhibit virion attachment to CX3CR1.

The chemokine domain of CX3CL1 (known as fractalkine) bears a CX3C motif (Fig. 5a) responsible for stimulating cytokine production following interaction with CX3CR1[31]. If maoto components interacted with the CX3C motif of fractalkine, maoto might also inhibit the attachment of fractalkine to CX3XR1. Because A549 cells do not produce interleukin (IL)-6 in response to fractalkine alone, cells were first primed with phorbol 12-myristate 13-acetate (PMA) and ionomycin, then incubated with fractalkine for 20 h. Intracellular IL-6 mRNA levels were significantly increased in A549 cells treated with fractalkine, but no increase was observed when maoto was added simultaneously (Fig. 5b). The interaction between the CX3C motif of fractalkine and maoto was confirmed by SPR (Fig. 5c). These results further suggested that binding of maoto to the CX3C motif of RSV G protein mediates anti-RSV activity.

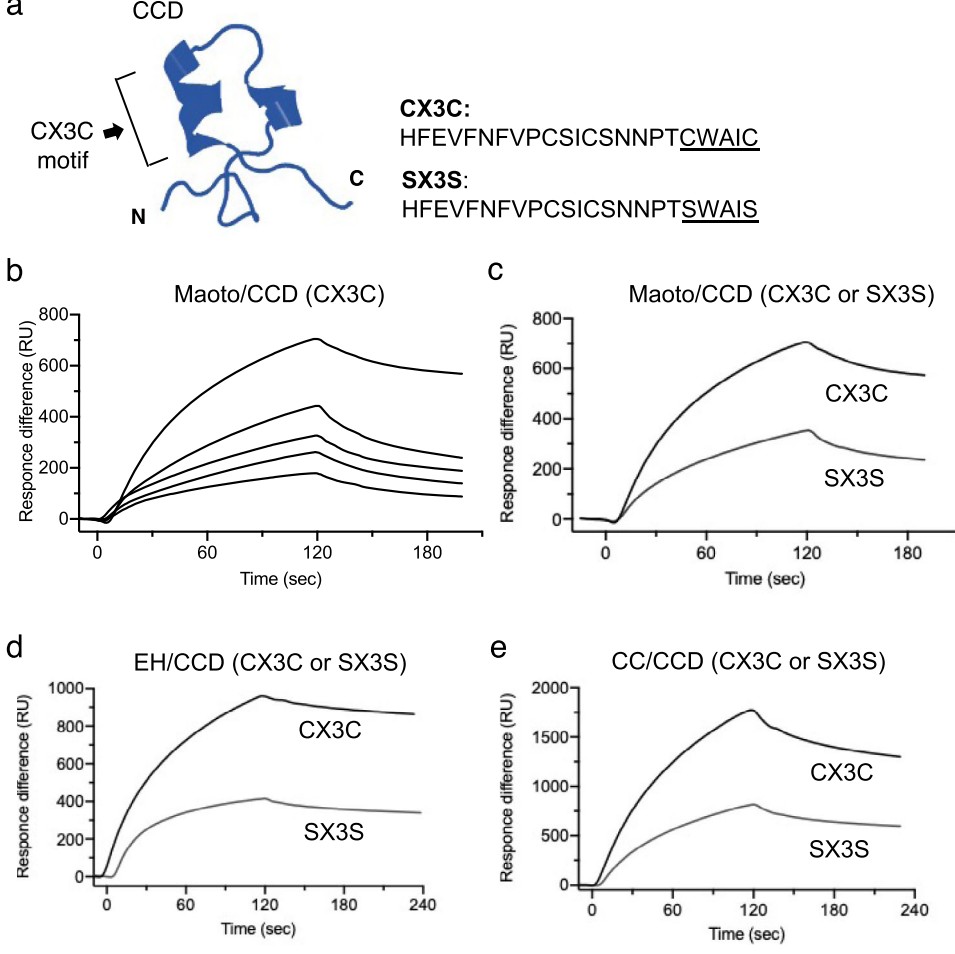

**Fig. 4 Specific binding of maoto to CX3C on the RSV G protein CCD. a** 3D-structural model of the RSV G protein CCD. The right column shows the amino acid sequence of CCD (aa 164–186). CX3C: CCD with CX3C motif (underline). SX3S: CCD with defective CX3C (Cys → Ser). **b** Binding of maoto to CX3C immobilized on a sensor chip by SPR. Five different concentrations (200, 300, 400, 500 and 600 μg/mL) of maoto were injected. **c** Binding of maoto (600 μg/mL) to CCD peptides (CX3C or SX3S) immobilized on a sensor chip by SPR. **d** Binding of EH (100 μg/mL) to CCD peptides immobilized on a sensor chip by SPR. **e** Binding of CC (100 μg/mL) to CCD peptides immobilized on a sensor chip by SPR.

**Reduction of viral loads by maoto in mice infected with RSV.** To confirm the antiviral effect of maoto in vivo, mice were intranasally infected with RSV and orally administered maoto, water, or prednisolone for 5 days (Experiment 1). No mice died following experimental infection. Bronchoalveolar lavage fluid (BALF) was collected 5 days post-inoculation, and viral loads were examined by PFU assay (Fig. 6a). Viral titers were remarkably reduced in maoto-treated mice compared with water- and prednisolone-treated control mice. In the additional in vivo study (Experiment 2), we compared the antiviral effect of maoto to a well-established comparator, palivizumab (PMB), resulting that maoto remarkably reduced the viral loads as well as PMB. There was no significant difference between the two groups (Fig. 6c). We also assessed whether AS and GR, which were not effective in the in vitro infection model, have the capacity to inhibit viruses. We found that AS/GR had mild efficacy to reduce viral loads in murine lung, but it is much less than maoto or PMB. Histopathological analysis of the lungs of water-, prednisolone-, and AS/GR-treated mice showed infiltration of mononuclear cells in the interstitial and perivascular regions (Fig. 6b, d). In contrast, lung sections from maoto- and PMB-treated mice revealed less infiltration of inflammatory cells. Because RSV induces inflammatory responses in the lung partly via the chemokine receptor, CX3CR1, we assessed levels of the inflammatory cytokines, IL-1β and IL-6, in the murine lung. We found that lung IL-1β and IL-6 mRNA levels in maoto-treated mice (2 g/kg) were decreased (Supplementary Fig. 7a, b). These results suggested that lung injury following RSV infection may be reduced by administration of maoto. Thus, the antiviral activity of maoto in vitro was confirmed by the in vivo experiments.

## Discussion

RSV infection in children has become a global issue for its high morbidity and mortality. The present study, showing that the traditional herbal medicine maoto has an antiviral activity against RSV, provides insights for the rational design of therapies against RSV infection. Maoto is a widely distributed drug in Japan, and its tolerability and effectiveness in patients with acute respiratory infections are widely accepted. Although maoto is composed of crude extracts, the clinical product is supplied as powders or granules, making it easy to take orally. Guidelines for quality management of the constitutive herbs of maoto are defined in the Japanese Pharmacopeia. Furthermore, the low cost of maoto provides an economic benefit for developing countries compared with antiviral monoclonal antibodies, which are costly and limited to intravenous administration. Therefore, repositioning and clinical trials of maoto may be more feasible than developing other candidate drugs to treat RSV infection. We are currently designing a clinical protocol for treatment of RSV infection in children using maoto granules.

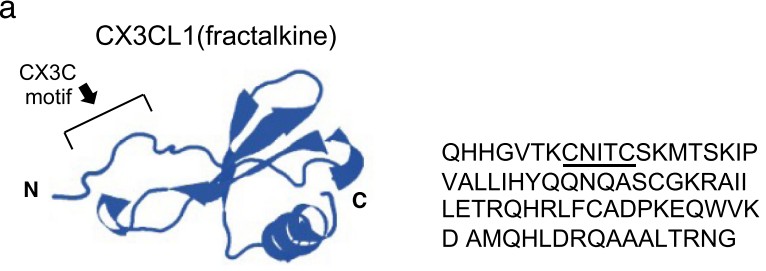

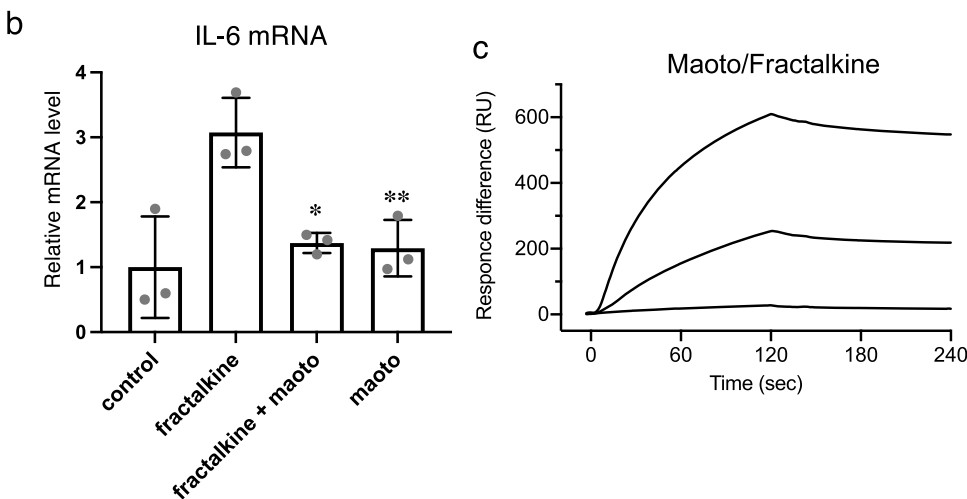

**Fig. 5 Interactions between maoto and the CX3C motif. a** 3D-structural model of the CXCL1 (fractalkine) chemokine domain. The right column shows the amino acid sequence of the fractalkine chemokine domain (aa 25–100). The CX3C motif is underlined. **b** Induction of IL-6 by fractalkine. A549 cells were primed with PMA (50 ng/mL) and ionomycin (1 μg/mL) for 4 h, and then treated with fractalkine (100 nM) in the presence or absence of maoto (25 μg/mL) for 20 h. Intracellular IL-6 mRNA levels were measured by RT-PCR. The results show individual values and mean ± error bars SD (n = 3). One-way ANOVA followed by Dunnett's post-test. *P < 0.05 and **P < 0.01 versus fractalkine alone. **c** Interaction of maoto with fractalkine immobilized on a sensor chip. Binding was assessed by SPR. Three different concentrations (20, 100 and 200 μg/mL) of maoto were injected.

The epitope in RSV G protein, CCD, responsible for attachment to host cells has recently been targeted by therapeutic and prophylactic monoclonal antibodies. However, poor antigenicity of the CCD and high development costs have made it difficult to develop effective neutralizing antibodies for clinical use. Both maoto and novel antibodies against the CCD may inhibit the infectivity and inflammatory activity of RSV, as they were thought to block attachment of CCD CX3C motif to CX3CR1. At present, it is unclear whether EH and CC in maoto interact preferentially with the CCD CX3C motif or with broader sites including the CX3C motif. It is also possible that maoto may interact with the RSV F protein or the heparin-binding domain immediately adjacent to the CX3C motif of CCD; the former fuses to host cell membrane for uncoating and the latter plays an important role in attachment to some kinds of cultured cells[32]. Our preliminary experiments by SPR showed that EH and CC could also interact with F protein. If maoto can interact with multiple viral protein components, it would broaden the spectrum of targets acted on by maoto to inhibit RSV attachment, and it may help prevent drug resistance. The anti-RSV effect of maoto was mediated by EH and CC, each containing hundreds of pharmacologically active compounds, and it remains possible that multiple compounds interact with the RSV G protein CCD. We are working to identify the specific molecule(s) involved in the antiviral effect of EH and CC by liquid chromatography and mass spectrometry. Preliminary experiments suggest that EH has at least two compounds identified by the screening for antiviral

activity. According to the literatures, other natural compounds derived from plants were reported to have an anti-RSV activity[33–36]. As well as maoto, Baicalin from Scutellaria Baicalensis blocked RSV attachment, however the precise mechanisms in molecular levels are not elucidated[30].

The unique characteristic of traditional herbal medicine, called kampo, was marked by the multiple components with diverse functions. Maoto may include molecules for relieving symptoms as well as those for inhibiting virus infection. Of plants mixed in maoto, EH and AS had the bronchodilation capacity which is important for RSV-infected children with bronchial asthma[37]. In the present study, AS and GR have a less antiviral effect than EH and CC, while they have a capacity to reduce the IL-1b induction as well as maoto. We have recently reported that maoto had a clinical effectiveness for seasonal influenza without inferiority to standard neuraminidase inhibitors. The anti-viral mechanism against influenza is that EH and CC in maoto inhibited the endosomal acidification, resulting in the inhibition of virus uncoating[29]. Interestingly, this mechanism is entirely different from the anti-viral mechanism against RSV infection in the present study. The effective concentration of maoto in the culture system of the present study was ten times less than that of influenza study. However, we also found that increasing dose of maoto could inhibit the propagation of RSV RNA even in the entry or post-uncoating phase (Supplementary Fig. 8), suggesting that host defense mechanisms or another antiviral mechanisms were induced when cells were treated with high concentration of

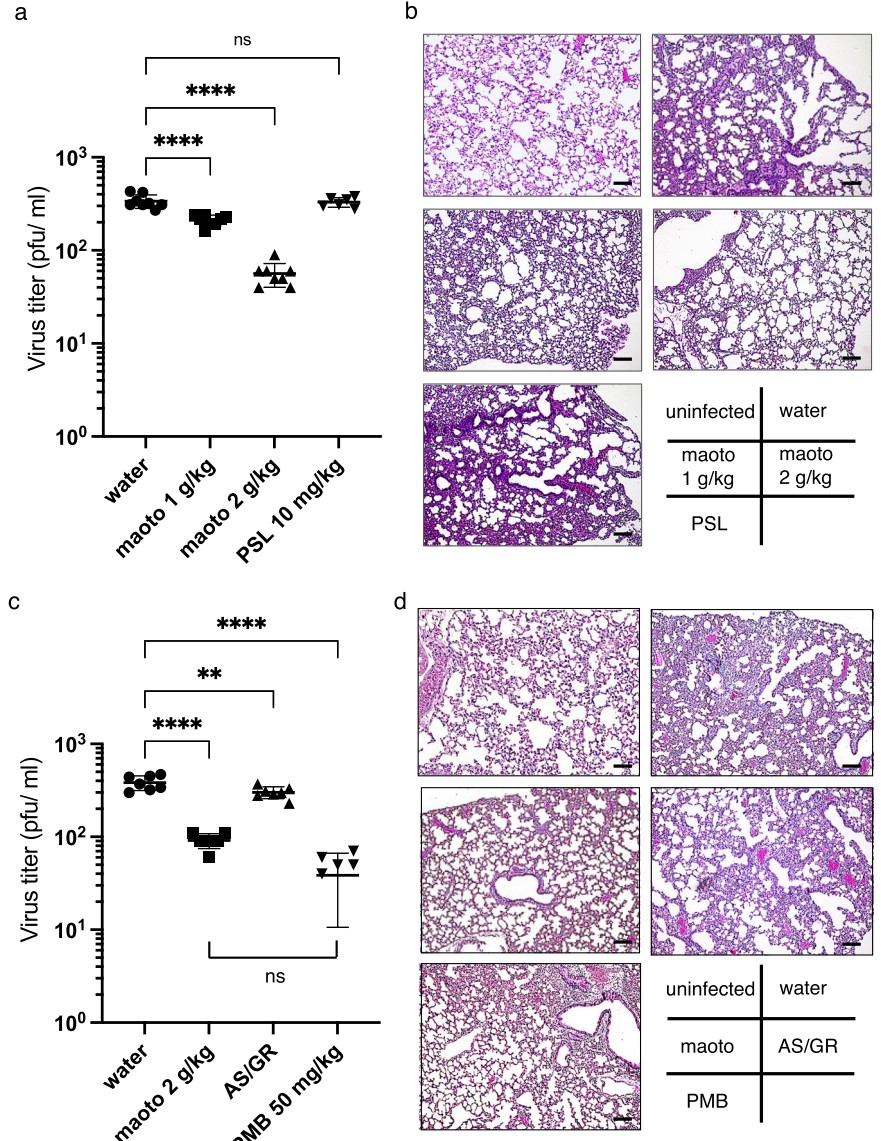

**Fig. 6 Reduction of viral load by maoto in mice infected with RSV. a, b** Experiment 1. **c, d** Experiment 2. **a** Female BALB/c mice were intranasally infected with RSV (1.8 × 10[6] PFU), and then orally administered maoto (1,000 mg/kg or 2,000 mg/kg), water (10 mL/kg), or prednisolone (10 mg/kg) for 5 days. Viral loads in BALF determined by PFU assay. **b, d** Histopathological findings in the lungs of mice infected with RSV (x100). Sections were stained with haematoxylin and eosin. **c** Female BALB/c mice were intranasally infected with RSV (1.8 × 10[6] PFU), and then orally administered maoto (2,000 mg/kg), water (10 mL/kg), AS/GR (645 mg/kg, 194 mg/kg, respectively), or PMB (50 mg/kg) for 5 days. Viral loads in BALF determined by PFU assay. **a, c** show individual values and mean ± error bars SD (a: n = 8, c: n = 7). One-way ANOVA followed by Dunnett's post-test. **$P < 0.01$, ****$P < 0.0001$ and ns; not significant. PSL: prednisolone. AS: Armeniacae Semen, GR: Glycyrrhize Radix, PMB, palivizumab. **b, d** Scale bars indicate 50 μm.

maoto. Unlike the existing anti-virals, there may be several independent anti-viral mechanisms in maoto. We have already reported that maoto could inhibit the serum proinflammatory cytokine responses and prostaglandin E2 levels in rats after Poly I:C injection, which was the model of acute RNA virus infection[25]. Taken together, it is suggested that maoto has both anti-viral and anti-inflammatory activities.

Maoto is the clinical drug targeting the CCD of RSV G protein like monoclonal antibodies[5]. The mechanism is that EH and CC interact with CX3C motif on G protein CCD to block attachment to CX3CR1, the receptor for RSV G protein. The anti-viral effect of maoto was confirmed by the mouse infection model. The benefits of maoto are its low cost, ease of administration, and clinical track record in Japan. Considering the number of children

suffering from RSV infection in the world, drug repositioning and clinical trials of maoto are justified.

## Methods

**Cells and viruses**. A549 cell line derived from human lung cancer, and RSV-A2 and -B were purchased from ATCC (Manassas, Virginia, US). Vero cells were from lab stock. The cell lines were cultivated in D-MEM supplemented with 10% FBS and penicillin/streptomycin at 37 °C in 5% $CO_2$. Viruses were initially propagated in A549 cells. The supernatant of culture was titered by PFU assay, and stored frozen under −80 °C, as a stock. Titers of viruses from culture supernatants and bronchoalveolar lavage fluid (BALF) from lungs of RSV-infected mice were determined by conventional PFU assay[38]. Briefly, HEp-2 cells were inoculated with RSV in serially diluted medium, followed by the incubation for 60 min, and then, overlaid with 0.3% warm Noble agar (Merck, Darmstadt, Germany). HEp-2 cells were cultured for 6 days at 37 °C in 5% $CO_2$ condition. The cells were stained with

neutral red and viral titers were determined by counting plaques. PBS was used for the control.

**Antiviral reagents**. Maoto, provided by Tsumura (Tokyo, Japan), is an extracted mixture of four plants (Supplementary Table 1), Ephedrae Herba (ephedra; EH), Armeniacae Semen (apricot; AS), Cinnamomi Cortex (cinnamon; CC), and Glycyrrhize Radix (licorice; GR). Crude drug pieces from EH, AS, CC, and GR were decocted by boiling water at a ratio of 10:10:8:3, respectively. Maoto powder was obtained via concentration and spray-drying of the decoction. The quality management of maoto is confirmed to the Japanese Pharmacopeia. Before experimental use, maoto powder was dissolved and incubated in warm PBS for 1 h, and supernatant was collected after sedimentation at 3,000 xg, then filtered through a 0.45 μm membrane filter. Dissolved maoto were stored at −80 °C until use. As reported in the previous study, cytotoxicity of maoto on A549 cells was seen over 1000 μg/mL for 24 h.

**Real-time PCR analysis**. RNA extraction was carried out using ISOGEN II (Nippon Gene, Tokyo, Japan) and cDNAs were synthesized with a prime script RT reagent kit with gDNA eraser (Toyobo, Osaka, Japan), according to the manufacturer's instructions. Real-time PCR analysis was performed using SYBR Green Kit (Takara, Kusatsu, Shiga, Japan) with CFX connect real-time PCR detection system (Bio-Rad; Hercules, California, US) by 40 cycles of 95 °C for 15 s and 60 °C for 1 min, and analyzed with CFX 3.1 software (Bio-Rad). The measured viral RNA is plotted on a linear scale as a percentage inhibition compared to the control in each figure. The sequences of primers are listed below (forward and revers):

human GAPDH: GCACCGTCAAGGCTGAGAAC and ATGGTGGTGAAGA CGCCAGT

RSV-A (N region): CATCCAGCAAATACACCATCCA and TTCTGCACAT CATAATTAGGAGTATCAA

RSV-B (P region): ACGCTACAAGGGCCTCATAC and TGCAATGCCAAA GTGCACAA

mouse IL-1β: CCTTCCAGGATGAGGACATGA and TGAGTCACAGAGGA TGGGCTC

mouse IL-6: GAGGATACCACTCCCAACAGACC and AAGTGCATCATCGT TGTTCATACA

human IL-6: ACTCACCTCTTCAGAACGAATTG and CCATCTTTGGAAG GTTCAGGTTG

**Western blotting analysis**. Cells were harvested in a lysis buffer (100 mM Tris-HCl (pH 6.8), 2% SDS, 20% glycerol, 2% β-mercaptoethanol, and 0.4 mg/ml bromophenol blue), and boiled at 100 °C for 10 min. Soluble proteins (25 μg) were separated by SDS-PAGE and transferred to a PVDF membrane (Bio-Rad) by Trans-blot turbo (Bio-Rad). The membrane was immersed with the Can Get Signal PVDF blocking Reagent (Toyobo) for 1 h, and incubated with monoclonal antibodies against RSV-G protein (1:1000, 94966; Abcam, Cambridge, UK) and β-actin (1:5000, A5316; Sigma-Aldrich, St. Louis, Missouri, US) for overnight. After washing with PBS containing 0.1% tween 20, the membrane was blotted with horseradish peroxidase-conjugated secondary antibody followed by visualization with a chemiluminescence agent, ECL Prime Western Blotting Detection Reagent (GE Healthcare, Chicago, Illinois, US) by LAS-3000 (GE Healthcare).

**Immunofluorescence confocal microscopy**. Cells were fixed with 4% paraformaldehyde at room temperatures for 15 min, and stained with primary antibody against RSV G protein (1:500, 94966; Abcam) and CXCR1 (1:500, 8021; Abcam), followed by further incubation with secondary anti-mouse antibody conjugated with Alexa Fluor 488 (1:1000, A11029; ThermoFisher, Waltham, Massachusetts, US). Samples were mounted by mounting agent (Prolong Diamond antifade mountant with DAPI; ThermoFisher) and observed under a confocal laser scanning microscopy LSM710 (Zeiss, Oberkochen, Germany). In some experiment (Figs. S4 and 5), fluorescein foci were counted by ImageJ software (Wayne Rasband, Bethesda, Maryland, US).

**Surface plasmon resonance (SPR)**. A Biacore J system (GE Healthcare) was used to study the physical interaction between G protein, CCD peptide, and fractalkine with maoto components. All experiments were conducted at 25 °C. Purified G protein (with sugar or without sugar) and CCD peptide from RSV strain A containing a C-terminal 6xHis tag were immobilized on a Ni-NTA sensor chip (GE Healthcare). Fractalkine was immobilized on a CM5 sensor chip (GE Healthcare). Samples of maoto were injected at a flow rate of 30 μl/min for association. Disassociation was performed over 120 s interval. The apparent affinity was evaluated with BIAevaluation software (GE Healthcare).

**Proximity ligation assay (PLA)**. PLA (Duolink; Sigma-Aldrich) was done according to the manufacture's instruction. Briefly, A549 cells were attached with RSV (MOI = 10) for one h on ice, and washed with cold PBS. The cells were fixed with 4% paraformaldehyde at room temperature for 15 min, followed by incubation with the blocking agent. The samples were incubated with primary antibodies

against G protein (1:500, 94966, Abcam) and CX3CR1 (1:500, 8021, Abcam), followed by binding of the PLA probes. Circle DNA between the two probes was amplified with fluorescent-labeled oligonucleotides. The samples were analyzed by the confocal microscopy, LSM-710.

**Production of recombinant protein and peptide**. C-terminus 6x His-tagged non-glycosylated recombinant RSV G (aa 66–297) was produced and purified[39] as follows. Codon optimized RSV G gene fragment was purchased from GenScript (Piscataway, New Jersey, USA) and cloned into pET21a (+) vector in NdeI and HindIII restriction enzyme sites. Resulting vector was used for transformation of *E. coli* BL21(DE3) cells and the transformants were cultivated at 37 °C until OD600 reached to 0.3. Then, isopropyl β-D-thiogalactopyranoside was added and the cells were further cultivated for 6 h. The cells were harvested and resuspended in the lysis buffer and sonicated. After removal of cell debris, the proteins were precipitated with the saturated ammonium sulfate at 4 °C. The precipitant was resuspended in a high salt buffer (10 mM Tris-HCl, pH8.0, 1 M ammonium sulfate) and subjected to an hydrophobic column (Hitrap-Phenyl HP column, GE Healthcare) connecting to FPLC (AKTA Explorer 10, GE Healthcare). The proteins were eluted with 10 mM Tris-HCl pH 8.0, the eluate was diluted with a buffer (20 mM Tris HCl, pH 7.6 and 0.5 M NaCl) and applied to an affinity column (His-trap HP column, GE Healthcare) and bound proteins were eluted with a linear gradient (20mM-500mM imidazole) of the buffer. Purified fractions were confirmed by 15% SDS-PAGE followed by CBB staining, and stored at 4 °C. Glycosylated RSV G was purchased from (Sino Biological, Beijing, China). The CCD peptides with or without CX3C motif (aa 164–186) with 6xHis-tag at C-terminus were purchased from Cosmo Bio (Tokyo, Japan), and their sequences of amino acid residues are listed below.

CCD with CX3C motif: HFEVFNFVPCSICSNNPTCWAICHHHHHH
CCD without CX3C motif: HFEVFNFVPCSICSNNPTSWAISHHHHHH

**Stimulation of A549 cells by fractalkine**. A549 cells were stimulated with 50 ng/ml PMA (Merck) and 1 μg/ml ionomycin (Merck) for 4 h, washed, and followed by treatment with 100 nM fractalkine (Fujifilm, Tokyo, Japan) in the presence or absence of maoto (25 μg/ml) for 20 h. Cell lysates were assayed by RT-PCR for IL-6 mRNA.

**Experimental murine infection with RSV**. Female BALB/c mice were purchased from SLC (Hamamatsu, Shizuoka, Japan), and used for the experiment at 6 weeks old. For the RSV-A2 infection, RSV ($1.8 \times 10^6$ PFU/0.1 mL/mouse) was intranasally inoculated. In the experiment 1, maoto (1 or 2 g/10 mL/kg), prednisolone (10 mg/10 mL/kg) or distilled water (10 mL/kg) were given to the mice by oral gavage for 5 days from one h after RSV inoculation until 4 days post-inoculation ($N = 8$). In the experiment 2, maoto (1 or 2 g/kg), AS/GR mix (AS: 645 mg/kg, GR: 194 mg/kg) or distilled water (10 mL/kg) were also given to the mice by oral gavage for 5 days ($N = 7$). Palivizumab (50 mg/kg), purchased from AstraZeneca (Cambridge, UK), were injected intraperitoneally once just after the RSV inoculation at day 0. Mice were sacrificed at 5 days post-inoculation, and collected BALF and lung tissue. For BALF sampling, 0.5 mL saline were injected three times into the trachea and retrieved fluids. The left lung tissue was kept frozen in RNAlater (Thermo Fisher Scientific) and other left lung was fixed with 10% formaldehyde solution. Paraffin sections of fixed tissue specimens were stained with hematoxylin and eosin. IL-1β and IL-6 mRNA from 3 individual mice were evaluated by RT-PCR. All animal experiments were approved by the Laboratory Animal Committee of Nihon Bioresearch (Hashima, Gifu, Japan) and Tsumura & Co (Tokyo, Japan), and performed in accordance with guidelines for the conduct of animal experiments in ministry of health, labor and welfare, Japan.

**Statistics and reproducibility**. Results were shown as mean value ± standard deviation (error bars) from at least three independent experiments. The statistically significant differences of 3 or more groups were determined by ANOVA with Dunnett's post test, and those of two group were determined by paired/ unpaired Student's *t* test or by Mann-Whitney *U* test, if the parameters were not distributed normally. A measure of IC50 was determined using logistic regression testing. By convention, ****$p$-values < 0.001, ***$p$-values < 0.001, **$p$-values < 0.01, *and* *$p$-values < 0.05. Data were analyzed with GraphPad Prism software (GraphPad Software, San Diego, California, US).

**Reporting summary**. Further information on research design is available in the Nature Research Reporting Summary linked to this article.

## Data availability

The data underlying figures presented in this study are available within the paper and Supplementary Data 1 and 2.

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

## Acknowledgements

We thank N. Okabe from the Life Science and Environment Research Centre for his valuable advice and Dr S. Ishino from Kyushu University for help with SPR experiments. We also thank Drs K. Ishii, R. Itoh, B. Chou, Y. Kurihara, and A. Shimizu from Fukuoka University, and Drs K. Oka, K. Ogata, and M. Hidaka from Fukuoka Dental College for technical advice. The structural model of CCD was generated using Phyre2 software (http://www.sbg.bio.ic.ac.uk/phyre2/html/page.cgi?id=help). The animal experiment was conducted by Nihon Bioresearch, Inc. This work was supported in part by a grant-in-aid for scientific research from the Japan Society for the Promotion of Science (19K07881).

## Author contributions

A.F. performed all aspects of this study. A.S. helped cell cultures and production of proteins. R.F. performed confocal microscopy, PLA experiments, and helped production of recombinant proteins. A.N. planned and conducted mouse experiments, and provided reagents. Y.I. helped to perform SPR experiments. K.H. helped to conceive the study and provided expertise. S.N. conceived and coordinated the study, and helped to write the manuscript. All authors read and approved final manuscript.

## Competing interests

A.N. is employed by Tsumura & Co, and S.N. has financial interests in Tsumura & Co. relevant to this research. The remaining authors declare no competing interests.
