## [Transparent Peer Review File · Communications Biology]

Reviewers' comments:

Reviewer #1 (Remarks to the Author):

Fujikane et al. demonstrated knockdown of RSV both in vitro and in vivo in a mouse model of RSV infection. Furthermore, they showed that this knockdown was mediated by an interaction of the Ephedrae Herba and Cinnamomi Cortex components of maoto with the G protein. Specifically, they demonstrate that these components bind to the CCD of the G protein, and that this binding can be attenuated through mutations in the CCD. The authors use a combination of high quality and varied assays to assert their claims, and the data seem convincing when taken together. This work represents a novel finding in the field, however there are some important aspects missing:

Specific comments:

1. Throughout the manuscript, t-test or Mann Whitney U are used between all groups and the control for statistical hypotheses comparison. This is appropriate in some of the experiments, but not in others. Specifically, either an ANOVA (or its nonparametric counterpart, as appropriate) or other multiple comparison correction should be used for Fig. 1a, c and d, Fig 2a, Fig 5b, Fig 6a, Fig S4, Fig S5, and Fig S6. For Fig. 1b and other dose response experiments with 4 or more measured doses, consider a regression analysis. The precise methodology used for each plot would be helpful.
2. Specifically for Fig. 1b and Fig S2a, a measure of neutralizing IC50 should be determined using logistic regression – testing of further dilutions of maoto may be required to accurately assess this.
3. The authors claim that incubating RSV with maoto prior to infecting cells reduces the infectivity of the virus (Fig. 2a). However, this effect could be due to the maoto present in the inoculum during infection such as in the experiments in Fig. 1b-d. To assess this, virus preps should be filtered after incubation with maoto to remove unbound maoto, such as with a 0.45 µm centrifugal filter, and the viral fraction measured against a control prep (see DOI 10.1021/nn405998v).
4. The in vivo experiment is compelling, however viral RNA load in either the lung or BAL should be measured to determine if viral knockdown is limited to complete virion or not.
5. Finally, a control extract (Armeniaca Semen or Glycyrrhize Radix) should be used in mice in the same dose range as maoto (1-2g/kg) to control for oral delivery of a similar dosage.

Minor comments:

1. The PLA experiments in Fig. 3g are convincing, especially in light of the quantification in Fig. S5. However, the microscopy images are hard to see since PLA generates small diffraction-limited spots. A cropped image of approximately a single cell that is blown up should be included to better detail the PLA signal for each condition.
2. Line 74, "the further clinical application..." should just be "further clinical application..."
3. Line 104, the dash after RSV-A2 should be removed.
4. Line 154, the dash between RSV and G should be removed
5. Line 215, "... activity of RSV. They were thought (sp) to block..." should be activity of RSV, as they were thought to block..."
6. Line 221, the sentence that start with "If maoto can interact with multiple viral components..." is unclear if it is referring to additional RSV strains or protein components - please clarify. Furthermore, the authors should indicate if the SPR between EH and CC are published elsewhere or if this is referring to unpublished preliminary results.
7. Indicate in the methods that drugs were given to the mice by oral gavage rather than feeding. While apparent when noting the dosing in this section, this will improve clarity for readers.

Reviewer #2 (Remarks to the Author):

Fujikane et al. provide a study of the herbal medicine maoto and its effect on respiratory syncytial virus infection. The authors employ cell culture virus attachment and infectivity assays to assess the effects of maoto. Further, the experiments assess extracts from four maoto component herbs and focus on the effects of extracts from Ephedrae herba and Cinnamomo cortex. Virus attachment and the effect of maoto are measured indirectly by nucleic acid quantification, western blot and immunofluorescence visualization. The effect on RSV G is examined by surface plasma resonance experiments using RSV G protein purified from recombinant source as well as a short peptide composed of the purported region spanning the CX3C motif. The authors lastly assess the effect of orally administered maoto in vivo in mice intranasally infected with RSV.

The experiments and more specifically the interpretation of their results are challenging given that the active molecular entity and its molecular formula or molecular weight is not defined. This may be all the more important since maoto is claimed to affect different viruses and targets and the eventual deconvolution may help the scientific assessment of these effects.

Specific comments:

1. The assessment of the anti-RSV activity of maoto may benefit from clarification of the experimental "control" and perhaps the inclusion of additional controls. In figures 1, 2, S1, S2, S3 and S7, the binding of RSV is determined by recovery of RSV nucleic acid through a reverse-transcriptase real-time PCR assay. The quantification of signal in this assay is variable. It is therefore important that the authors show the variability in all experimental samples. The variability in "controls", is as important to include in the figures, as the variability of the maoto treated samples. In the submitted figures, all controls and all replicates are shown with 100% without error or standard deviation. It is unlikely that the qPCR assay results have no variation in the control and only showed variation in the treated samples. These data should be re-assessed plotting the variation of the control with the appropriate statistical analyses.
2. Further to point 1 above, inhibition of virus and quantification of viral nucleic acid is generally plotted on a log scale. The authors should consider plotting the analysis of vRNA on a log scale rather than linear scale as a percentage. The composition of the control is not defined. Is control simply PBS or an inactive herbal extract lacking EH and CC components?
3. The quantitative effect of maoto on RSV infection is not entirely convincing. Figure 2a and figure 2c show a significant effect on vRNA with 25 ug/mL of maoto and the effect with 25 ug/mL does not appear to differ relative to the 250 ug/mL concentration. In contrast, figure 2d shows no apparent antiviral effect with 25 ug/mL maoto, when analysing RSV protein level. The authors should address this discordance.
4. Association with recombinant RSV G protein is examined through surface plasma resonance experiments. In figure 3, the association phase of the reactions appear linear and it is not apparent whether binding to immobilized RSV G has reached equilibrium. Shouldn't there be a curvature with increasing ligand? Is the association/dissociation specific? Is there a difference in the association/dissociation with the unglycosylated G? If so, why?
5. The SPR data in figure 4 are qualitatively and quantitatively different from figure 3. The association phase has an apparent curvature lacking in the SPR experiments in figure 4. Are the authors measuring the same binding? Notably, the quantities of the herbal extract used in figure 4 (range 200-600 ug/mL) are 15-40 times higher than the quantities used in figure 3 (range 5- 40ug/mL).
6. The immobilized peptide with the 22 amino acid CX3C motif used in figure 4 has 4 cysteine residues, whereas the "defective" peptide has 2 cysteine residues. Have the authors discounted the possibility of non-specific reactive sulfide bond formation with components of the herbal extract that may account for these results?

7. The PLA assay in figure 3g shows green foci signals at both the 100 ug/mL and 250 ug/mL maoto that appear greater than no RSV infection control, and which may be marginally less than the 0 ug/mL maoto RSV infected comparator. The magnitude of the effect in this figure is inconsistent with the purported 250 ug/mL level depicted in figures 1 and 2.

8. The antiviral data in figure 6 are plotted on a linear scale. In vivo pharmacological assessment in the murine RSV challenge study should be plotted on a log scale and include a well-established comparator, such as palivizumab that is dosed parenterally. In vivo pharmacological assessment of a direct acting antiviral generally includes corresponding pharmacokinetic data showing plasma and target organ levels of the active pharmaceutical ingredient. The authors may wish to consider conventionally accepted research criteria for antiviral therapeutics and how they may address such norms in revised studies.

We wish to express our appreciation to the reviewers for their insightful comments that have helped us significantly improve the paper. We are very sorry to have extended the time limit of our response due to the re-examination of the animal study. In the following sections, you will find our response to each of your questions and suggestions. We are grateful for the time and energy you expended on our behalf.

Responses to reviewers

Response to the comments of reviewer #1

Specific comments

Q1. Specifically, either an ANOVA (or its nonparametric counterpart, as appropriate) or other multiple comparison correction should be used for Fig. 1a, c and d, Fig 2a, Fig 5b, Fig 6a, Fig S4, Fig S5, and Fig S6. For Fig. 1b and other dose response experiments with 4 or more measured doses, consider a regression analysis. The precise methodology used for each plot would be helpful.

A1. The reviewer's comment is correct. We again analyzed the data by ANOVA for Fig. 1a, c and d, Fig 2a, Fig 5b, Fig 6a, Fig S4, Fig S5, Fig S6 and Fig S7. P values (asterisks) are shown in each figure, and the methodology is written in each legend and the Materials and Methods section (Statistical analysis, red letters). For Fig 1b and Fig S2a, please see below.

Q2. Specifically for Fig 1b and Fig S2a, a measure of neutralizing IC50 should be determined using logistic regression testing of further dilutions of maoto may be required to accurately assess this.

A2. We re-examined the dose-dependent antiviral effect of maoto and determined the IC50 using logistic regression analysis. Figs. 1b and S2a were replaced by revised figures. We added an explanation about IC50 to the legends of Figs 1b and S2a and in the Result section (*Inhibitory effects of maoto in cultured cells infected with RSV*, red letters).

Q3. The authors claim that incubating RSV with maoto prior to infecting cells reduces the infectivity of the virus (Fig. 2a). However, this effect could be due to the maoto present in the inoculum during infection such as in the experiments in Fig. 1b-d. To assess this, virus preps should be filtered after incubation with maoto to remove unbound maoto, such as with a 0.45 μm centrifugal filter, and the viral fraction measured against a control prep.

A3. The reviewer has raised an important question. Indeed, when virus is exposed to 250 $\mu\text{g/ml}$ maoto, the remaining maoto (6.3 $\mu\text{g/ml}$) during the infection may influence the result according to the IC50 curve (revised Fig.1b). In order to deduce maoto in inoculum, we did an additional analysis without maoto (control) and with 25 $\mu\text{g/ml}$ maoto. The remaining maoto during the infection (< 0.6 $\mu\text{g/ml}$) reduced the infectivity by only 20% (see Fig. 1b), which is lower than the 25 $\mu\text{g/m}$ shown in the original results (60%). The data with 250 $\mu\text{g/ml}$ maoto was

deleted. The reviewer suggested removing the unbound maoto by filtration. Because, in the present study, maoto extract was passed through a 0.45 μm filter before the experiment, unbound maoto could not be removed by filtration. We replaced the former Fig 2a with a revised one, and accordingly changed the figure legend (red letters).

Q4. The in vivo experiment is compelling, however viral RNA load in either the lung or BAL should be measured to determine if viral knockdown is limited to complete virion or not.

A4. We are interested in the comment above. Accordingly, we performed an additional in vivo experiment and extracted whole RNA from the lung. However, we were not able to obtain reliable and stable results by the standard real-time PCR method due to the abundant inhibitory cellular RNAs compared to the lesser viral RNAs. We have tried PCR many times with various conditions of experiments using virus-specific primers referring to the relevant literature for the measurement of RSV RNA from the infected lung (Bio Protoc. May20;6(10), 2016), but all attempts were unsuccessful. Finally, we were compelled to abandon the measurement of viral RNA from the lung. We consider that in situ hybridization or another technology may help us measure it in future experiments. Instead of this, we have reconfirmed the former data by this additional work through measuring the virus titer in BALF by pfu assay. Unfortunately, we could not measure viral RNA in BALF, because the fluid of BAL, a small amount, was all needed for the pfu assay.

Q5. Finally, a control extract (Armeniacae Semen or Glycyrrhize Radix) should be used in mice in the same dose range as maoto (1-2g/kg) to control for oral delivery of a similar dosage.

A5. The comment above is very important. We had no idea about the effectiveness in vivo of Armeniacae Semen (AS) and Glycyrrhize Radix (GR), which were not effective in the in vitro infection model. Accordingly, we did an additional in vivo experiment with AS and GR in the same dose range as maoto (2g/kg), which were mixed and delivered to mice through oral syringe for five days. We have found that AS/GR have mild efficacy to reduce viral loads in murine lung, but it is much less than maoto or palivizumab. We have confirmed that the responsible crude drugs are mainly EH and CC in vivo. Please allow us to add the revised data in Fig. 6, in which we also show the histopathological findings of the additional in vivo experiment. We found that maoto and palivizumab reduced the infiltration of inflammatory cells, but AS/GR did not.

We revised the IL-1 and IL-6 mRNA production in the mouse lung. Interestingly, we found that AS/GR reduced the IL-1 mRNA production as well as maoto and palivizumab, regardless of a lesser antiviral effect. We would like to add this result to Fig. S6.

Minor comments

1. The reviewer asked to clarify the PLA signals in the microscopy images in Fig 3g. We now show a cropped image of few cells in each photo.
2. Line 74, “the further clinical application...” is changed to “further clinical application...”
3. Line 104, the dash after RSV-A2 is removed.

4. Line 154, the dash between RSV and G is removed.

5. Line 215, "... activity of RSV. They were thought (sp) to block..." is changed to "activity of RSV, as they were thought to block..."

6. According to the reviewer's comment, we have changed the sentence as below (Line 221).

Before: "If maoto can interact with multiple viral components..."

After: "If maoto can interact with multiple viral protein components..."

Next, the reviewer stated we should indicate if the SPR are published elsewhere. The SPR results are our preliminary data as described in Line 221.

7. The reviewer asked whether drugs were given to mice by oral gavage. The way of drug delivery is oral gavage by syringe. This sentence is added to the Materials and Methods section (*Experimental murine infection with RSV*) and the legend of Fig. 6 (red letters).

Response to the comments of reviewer #2

Specific comments

Q1. The assessment of the anti-RSV activity of maoto may benefit from clarification of the experimental "control" and perhaps the inclusion of additional controls. In figures 1, 2, S1, S2, S3 and S7, the binding of RSV is determined by recovery of RSV nucleic acid through a reverse-transcriptase real-time PCR assay. The quantification of signal in this assay is variable. It is therefore important that the authors show the variability in all experimental samples. The variability in "controls", is as important to include in the figures, as the variability of the maoto treated samples. In the submitted figures, all controls and all replicates are shown with 100% without error or standard deviation. It is unlikely that the qPCR assay results have no variation in the control and only showed variation in the treated samples. These data should be re-assessed plotting the variation of the control with the appropriate statistical analyses.

A1. We appreciate the reviewer's comment on the "controls" variability in the RT-PCR assay. Accordingly, we have re-assessed the data and plotted controls with variability (SD) in Figs. 1, 2, S1, S2, S3 and S7, along with the appropriate statistical analyses, ANOVA and Student's *t* test. The revised method of statistical analysis, which was also pointed out by Reviewer #1, was added to the the Materials and Methods section (*Statistical analysis*, red letters).

Q2. Further to point 1 above, inhibition of virus and quantification of viral nucleic acid is generally plotted on a log scale. The authors should consider plotting the analysis of vRNA on a log scale rather than linear scale as a percentage. The composition of the control is not defined. Is control simply PBS or an inactive herbal extract lacking EH and CC components?

A2. There is a reason we plotted the data of vRNA on a linear scale as a percentage. Actually, if the proliferated

vRNA in cultured cells is measured in comparison with vRNA immediately after inoculation into cells (the ‘base value’), the Y axis is generally plotted on a log scale as a fold-increase to the base value. In the present study, we cannot compare vRNA to the base value, because viruses cannot go into cells if maoto exists in media. For that reason, we plot the untreated control vRNA after proliferation as 100%, namely, each vRNA value is expressed by an inhibition ratio. We hope you can accept our use of this particular method in our study. We have addressed this issue to inform the readers of our method in *Real-time PCR analysis* of the Materials and Methods section (*Real-time PCR analysis*) and each figure legend (red letters).

We were asked to define the composition of the control. It is simply PBS, which has been reflected in the Materials and methods section (*Cells and Viruses*) and each figure legend (red letters).

Q3. The quantitative effect of maoto on RSV infection is not entirely convincing. Figure 2a and figure 2c show a significant effect on vRNA with 25 ug/mL of maoto and the effect with 25 ug/mL does not appear to differ relative to the 250 ug/mL concentration. In contrast, figure 2d shows no apparent antiviral effect with 25 ug/mL maoto, when analysing RSV protein level. The authors should address this discordance.

A3. We think the reviewer’s comment is important. Figure 2d visualizes the virus G protein on the cell surface by western blot analysis, which is less sensitive than PCR. To overcome this, we had to inoculate more viruses at a moi of 10 (usually 1), which needs more maoto concentration (>100µg/ml) to inhibit the virus attachment to the cell surface. In the revised study, we were able to improve this discordance by changing the culturing system, inoculating viruses at a moi of 1 onto a much larger scale of cells using a large culturing plate. We found that the antiviral effect of maoto was shown at a concentration 25 µg/ml, consistent with other experiments. This result has been reflected in Fig. 2d and the figure legend.

Q4. Association with recombinant RSV G protein is examined through surface plasma resonance experiments. In figure 3, the association phase of the reactions appears linear and it is not apparent whether binding to immobilized RSV G has reached equilibrium. Shouldn’t there be a curvature with increasing ligand? Is the association/dissociation specific? Is there a difference in the association/dissociation with the unglycosylated G? If so, why?

A4. Our experiment shown in Figure 3 indicated the result of SPR, in which maoto was loaded onto a RSV G-immobilized sensor chip. The maoto components were also examined. Our sensorgram showed linear responses for the association phase and no dissociation phase, as the reviewer pointed out. The analytes used in this experiment were the mixture of the maoto components, but not any single molecule. That is why multiple components bind to some compound binding sites of the G protein and no longer dissociate. The binding may include some non-specific binding. However, the sensorgrams presented were the results from the subtraction of the responses for a non-immobilized chip, and therefore, we believe, at least some components in maoto have binding ability to the G protein immobilized on the sensor chip. The recombinant RSV G protein produced in *E. coli* was not glycosylated. We used this recombinant protein as the ligand instead of the glycosylated G protein. Comparison of Figure 3a and

3G suggested that glycosylation is not critical for binding of a maoto component, as described in the Text. The analytes were the mixture, but not an isolated single compound, in all of the SPR in Figure 3. The shape of the sensorgrams obtained by this work is often observed in cases where this mixture is used. The K_D value cannot be precisely calculated.

Q5. The SPR data in figure 4 are qualitatively and quantitatively different from figure 3. The association phase has an apparent curvature lacking in the SPR experiments in figure 4. Are the authors measuring the same binding? Notably, the quantities of the herbal extract used in figure 4 (range 200-600 ug/mL) are 15-40 times higher than the quantities used in figure 3 (range 5- 40ug/mL).

A5. This is an important question. We consider that the components in maoto that interact with G protein are a mixture. Actually, in the preliminary experiment, we found that at least 4 components from maoto extract interacted with G protein, and further research may identify other components. The molecular size of G protein is much larger than the CCD peptide, and G protein may include more binding sites for herbal components than the CCD peptide. Therefore, we think that SPR analysis with CCD peptide may require higher concentrations of maoto than SPR using G protein to detect the association because active components interacting with CCD peptides become reduced. Such cases are often observed for the in vitro binding analysis.

Q6. The immobilized peptide with the 22 amino acid CX3C motif used in figure 4 has 4 cysteine residues, whereas the “defective” peptide has 2 cysteine residues. Have the authors discounted the possibility of non-specific reactive sulfide bond formation with components of the herbal extract that may account for these results?

A6. In the course of the research, we used a mass spectrometer to determine if the CCD peptide had disulfide bonds. Also, there were no compounds covalently bound to the cysteine residues in our attempt to identify maoto components that interact with CCD peptides by MS (data not shown). Therefore, there is little possibility that free sulfide residues in CCD peptides affect the SPR results.

Q7. The PLA assay in figure 3g shows green foci signals at both the 100 ug/mL and 250 ug/mL maoto that appear greater than no RSV infection control, and which may be marginally less than the 0 ug/mL maoto RSV infected comparator. The magnitude of the effect in this figure is inconsistent with the purported 250 ug/mL level depicted in figures 1 and 2.

A7. We are very sorry that the Fig. 3g photos are not clear. This was also pointed out by Reviewer 1. Fig. 3g has been revised. Please note that PLA needs more viruses (moi=10) to visualize the foci, and needs more maoto concentration at $>100\mu\text{mg/ml}$. In addition, green foci signals at 100 ug/mL and 250 ug/mL maoto and uninfected control included nonspecific background signals, which was also shown by histogram in Fig.S5.

Q8. The antiviral data in figure 6 are plotted on a linear scale. In vivo pharmacological assessment in the murine

RSV challenge study should be plotted on a log scale and include a well-established comparator, such as palivizumab that is dosed parenterally. In vivo pharmacological assessment of a direct acting antiviral generally includes corresponding pharmacokinetic data showing plasma and target organ levels of the active pharmaceutical ingredient. The authors may wish to consider conventionally accepted research criteria for antiviral therapeutics and how they may address such norms in revised studies.

A8. According to the reviewer' comment, the antiviral data in fig. 6 has been plotted on a log scale. An additional murine RSV challenge study with a well-established comparator, palivizumab, has been performed referring to the report 'Antimicrob Agents Chemother. 48:5;1811, 2004', resulting in maoto remarkably reducing the viral titer in BALF as well as palivizumab. There was no significant difference between the two groups. The result has been added to the Result section (*Reduction of viral loads by maoto in mice infected with RSV*) and the supplemental figure (Fig. S8).

The reviewer asked us to show plasma and target organ levels of the active pharmaceutical ingredient. Unfortunately, such active compounds to block infectivity of virus have not been identified. We are now working to identify them in EH, and we may be able to show the compounds in the near future. We have already reported the pharmacokinetic data for representative compounds in maoto (Ref. 25). Known molecules, such as Ephedrine, Methyl ephedrine, Cinnamic acid, and Hippuric acid, were emerged in rat plasma at 1 to 8 hours after maoto treatment. If we can identify compounds that block the infectivity of RSV, we will assess the pharmacokinetic data showing plasma and target organ levels in mice.

For the last point, conventional antiviral therapeutics are categorized into purified small molecules, cytokines, and neutralizing antibodies. Herbal medicines are difficult to categorize because they include multifunctional ingredients, such as antiviral, anti-inflammatory, and vasoactive compounds. For instance, maoto includes antiviral compounds to at least RS and influenza virus and probably to mouse hepatitis virus (unpublished data) with anti-IL-1 and IL-6 functions. Additionally, the cost of herbal medicine is very low compared with newly designed antivirals. In the future, traditional herbal medicines may change the research criteria for antiviral therapeutics. Comments about the above have been added in the discussion section.

Again, thank you for giving us the opportunity to strengthen our study. We have worked hard to incorporate your feedback and hope that these revisions are sufficient make our paper acceptable for publication.

Sincerely,

Shigeki Nabeshima, MD, PhD

General Medicine,

Fukuoka University Hospital,

7-45-1 Nanakuma, Jonanku, Fukuoka, Japan
snabeshi@fukuoka-u.ac.jp

Reviewers' comments:

Reviewer #1 (Remarks to the Author):

My concerns were addressed adequately and appropriately. The added analyses and data has strengthened the conclusions made in the paper.

Reviewer #2 (Remarks to the Author):

Summary

Fujikane et al in a revised manuscript provide revised analyses of data from the original submission, as well as re-designed in vivo studies of the herbal medicine maoto and its effect on respiratory syncytial virus infection. The revised manuscript maintains organizational structure and experimental approaches that examine cell culture RSV attachment and infectivity to assess the inhibitory effects of maoto. Virus attachment and the effect of maoto are measured indirectly by nucleic acid quantification, western blot and immunofluorescence visualization. RSV G is proposed as a potential target and examined by surface plasma resonance experiments using RSV G protein purified from recombinant source, as well as a short peptide spanning the purported binding region spanning the CX3C motif of RSV G. The authors assess the effect of orally administered maoto in vivo in mice intranasally infected with RSV and present the results of a second in vivo mouse experiment with a benchmark positive control (palivizumab) comparator.

The revised manuscript's experiments and results, like the original, are a challenge to interpret given that the active molecular entity and its molecular formula or molecular weight is not defined. The authors do mention this unanswered gap in a section of the discussion where they suggest multiple mechanisms of action.

Specific comments on the revisions and revised manuscript:

1. In revised figures 1, 2, S1, S2, S3 and S7 the binding of RSV is determined by recovery of RSV nucleic acid through a reverse-transcriptase real-time PCR assay. The quantification of signal in this assay is variable and the authors now do show the variability in all experimental samples with a revised statistical analyses. The authors should indicate whether the multiple points represent inter- or -intra experimental replicates. Replicates from different experiments would be most appropriate.
2. Inhibition of virus and quantification of viral nucleic acid can be plotted on a log scale, even for viral entry inhibitors. The replot of a dose response by non-linear regression fitted to a Hill plot, is an improvement.
3. The apparent discordance in the quantitative effect of maoto on RSV infection (i.e. the original manuscript Figure 2a and figure 2c showed a significant effect on vRNA with 25 ug/mL of maoto, whereas figure 2d showed no apparent antiviral effect with 25 ug/mL maoto, when analysing RSV protein level) was addressed by the authors by noting that different MOIs were used in the different experiments. That different MOI impacts the potency is a property that should be investigated more thoroughly, over a broader range of MOIs.
4. Association with recombinant RSV G protein is examined through surface plasma resonance experiments. In figure 3, the association phase of the reactions appear linear and it is not apparent whether binding to immobilized RSV G has reached equilibrium and whether the association/dissociation is specific. The authors respond that the binding may include "some non-specific binding". Unfortunately, this response does not help the reader interpret the result, and may lead to unintended misinterpretation. Unless the authors can distinguish specific from non-specific binding, the SPR results are confusing and should be removed.
5. The SPR experiments with CCD peptide in figure 4 required quantities of the herbal extract up to 15-40 times higher than the quantities used in figure 3 (range 5- 40ug/mL) and the authors respond that G protein is larger than CCD peptide, to account for the quantitative differences. Since the

authors concede the SPR experiments may involve non-specific and specific binding, all of these SPR experiments should be removed, until the authors can identify and examine the specific components by SPR.

6. The authors address the possibility of non-specific reactive sulfide bond formation with components of the herbal extract by noting that MS analysis did not detect covalent binding to cysteines. If the SPR experiments are deleted, this clarification may not be required.

7. In the revisions and rebuttal, the authors indicate that the PLA assay in figure 3g, with quantitative discordance in the amount of maoto used relative to other figures, required a higher MOI infection condition. As mentioned in point 3 above, the authors should provide a better characterization of the MOI effect on the antiviral activity of maoto.

8. The antiviral in vivo pharmacological data in revised figure 6 is plotted on a log scale and includes the benchmark comparator palivizumab with well characterized antiviral effect (Antimicrob Agents Chemother. 48:5;1811, 2004). One notable aspect of the presented study design in experiments 1 and 2 of figure 6 is that the authors inoculate mice with 10^6 PFU/mouse, whereas the benchmark that they reference inoculated mice with 10^8 PFU/mouse. The authors use 100-fold less virus inoculum for their in vivo antiviral experiments relative to the reference study. Since the authors have observed that the MOI and amount virus impacts maoto cell culture potency, perhaps they may be concerned with using significantly less virus inoculum in their in vivo studies. The authors may want to provide an assessment, or explanation, regarding the potency of maoto when used in a model infected with 10^8 PFU/mouse. Lastly, the authors should specify which strain of RSV was used in the mouse studies.

"Inhibition of respiratory syncytial virus infectivity by Ephedrae Herba and Cinnamomi Cortex through interaction with G glycoprotein (#COMMSBIO-21-0462)"

Response to the comments of reviewer #2

We wish to express our appreciation to the reviewers for their insightful comments that have helped us significantly improve the paper. We also appreciate the time and effort you have dedicated to providing insightful feedback on ways to strengthen our paper. As Professor Si Ming Man, the editor, has thoughtfully grouped the comments of Reviewer 2, we will respond to them in that order.

1) *The reviewer asked you to state whether the data represent inter- or -intra experimental replicates.*

The in vitro data represent the inter-replicates. The data are mean values from at least three independent experiments, which are stated in the Materials and Methods section (red letters).

2) *The reviewer asked you to represent the “Inhibition of virus and quantification of viral nucleic acid” data on a log scale.*

Yes, we changed the way of expressing this from a linear to a log scale (Figs. 1a, 1d, 2a, 2c, S1a, S2b, S3, and S7).

3) *The reviewer asked you to examine the effect of maoto on vRNA versus viral protein using different MOI of the virus: We are happy for you to decide whether you want to provide this data, or if you feel that this is an issue which is beyond the scope of this manuscript.*

The reviewer commented “that different MOI impacts the potency is a property that should be investigated more thoroughly, over a broader range of MOI.” In response to that, we have done additional experiments that clarify the effect of maoto at a MOI of 10. As a result, the IC₅₀ of maoto by PCR has changed from 1.77 (MOI=1) to 21.57 µg/ml (MOI=10). As expected, an increasing number of viruses can weaken the effect of maoto both in protein and vRNA levels. We show this data in Fig. S8. The above sentence has been added to the Result section (red letters).

4) *The reviewer previously questioned the specificity of the SPR data and was not satisfied with your response that some non-specific binding may occur. The reviewer suggests removal of this data. Please consider whether you would like to keep or remove the data. If you decide to keep the data, you may want to make clear that “some non-specific binding may occur”.*

It is true that the binding affinity cannot be quantified precisely from the presented sensorgram. However, we subtracted background originated from the nonspecific binding of maoto components to the sensor chip without ligand. Therefore, the observed response of the sensorgrams in Fig. 3 showed a specific binding of maoto to the

RSV G protein, at least qualitatively. Then, we did a further SPR analysis using the CX3C peptide as the ligand to examine which part of the G protein can bind a maoto component. The sensorgrams in Fig. 4 clearly showed that the CX3C peptide bound a component of maoto. These sensorgrams are normal shapes indicating typical association/dissociation pattern. Unfortunately, the K_D value cannot be calculated because the concentration of the binding molecule in maoto is not known yet. Because we consider the SPR data to be the core of our study, we would like to keep them in our manuscript. We describe that our qualitative SPR data possibly contains non-specific binding in the Result section, according to the Editor's suggestion (red letters). We think the readers will understand our results of these binding experiments precisely and reviewer's worries can be escaped.

5) *The reviewer asked for an explanation on the potency of maoto when a higher dose of RSV is used, and that the strain of RSV used in the mouse studies should be stated.*

Reviewer 2 noted that we used 100-fold less virus inoculum for our in vivo antiviral experiments relative to the reference study (Antimicrob Agents Chemother. 48:5;1811, 2004). However, we think that the reviewer has misread the number of viruses in inoculation. The following text is a quote from the paper above (Materials and Methods section).

“BALB/c mice were intranasally inoculated once (on day 0) with $10^{7.5}$ to $10^{8.2}$ PFU of human RSV A2/ml in 100 μ l of 10% EMEM.”

Thus, the reference study used $10^{6.5}$ to $10^{7.2}$ PFU/mouse of RSV for inoculum, so the inoculated virus/mouse in the present study is 1.7 to 8.8-fold, not 100-fold, less than the reference paper. In addition, in another paper describing the effect of palivizumab on RSV-infected mice (BALB/c), the authors inoculated 2×10^6 PFU/mouse RSV (Human Gene Ther. 2021<https://doi.org/10.1089/hum.2021>). From the above, we think the viral number inoculated into mice in the present study is not so low. Please allow us to not change the explanation of the potency of maoto in our manuscript when a 100-fold higher dose of RSV is used.

We have described the strain of RSV (RSV-A2) both in the Materials and Methods section and the legend of Figure 6 according to the Editor's suggestion (red letters).

Again, thank you for giving us the opportunity to resubmit our study. We have revised the manuscript to incorporate your feedback to the best of our ability and hope that these revisions are sufficient to make our paper acceptable for publication.

Best regards,

Shigeki Nabeshima, MD, PhD,
General Medicine,
Fukuoka University Hospital, Japan